# Use of 3D Spheroid Models for the Assessment of RT Response in Head and Neck Cancer

**DOI:** 10.3390/ijms24043763

**Published:** 2023-02-13

**Authors:** Marilyn Wegge, Rüveyda Dok, Ludwig J. Dubois, Sandra Nuyts

**Affiliations:** 1Laboratory of Experimental Radiotherapy, Department of Oncology, KU Leuven, University of Leuven, 3000 Leuven, Belgium; 2The M-Lab, Department of Precision Medicine, GROW-School for Oncology and Reproduction, Maastricht University, 6229 ER Maastricht, The Netherlands; 3Department of Radiation Oncology, Leuven Cancer Institute, UZ Leuven, 3000 Leuven, Belgium

**Keywords:** head and neck cancer, human papillomavirus, radiotherapy, 3D spheroid models

## Abstract

Radiotherapy (RT) is a key player in the treatment of head and neck cancer (HNC). The RT response, however, is variable and influenced by multiple tumoral and tumor microenvironmental factors, such as human papillomavirus (HPV) infections and hypoxia. To investigate the biological mechanisms behind these variable responses, preclinical models are crucial. Up till now, 2D clonogenic and in vivo assays have remained the gold standard, although the popularity of 3D models is rising. In this study, we investigate the use of 3D spheroid models as a preclinical tool for radiobiological research by comparing the RT response of two HPV-positive and two HPV-negative HNC spheroid models to the RT response of their corresponding 2D and in vivo models. We demonstrate that HPV-positive spheroids keep their higher intrinsic radiosensitivity when compared to HPV-negative spheroids. A good correlation is found in the RT response between HPV-positive SCC154 and HPV-negative CAL27 spheroids and their respective xenografts. In addition, 3D spheroids are able to capture the heterogeneity of RT responses within HPV-positive and HPV-negative models. Moreover, we demonstrate the potential use of 3D spheroids in the study of the mechanisms underlying these RT responses in a spatial manner by whole-mount Ki-67 and pimonidazole staining. Overall, our results show that 3D spheroids are a promising model to assess the RT response in HNC.

## 1. Introduction

Head and neck cancer (HNC) is the sixth most common cancer and cause of cancer-related death worldwide, with more than 500,000 new cases and 380,000 deaths yearly [1]. Currently, around 75–85% of HNC cases are associated with tobacco and alcohol abuse, which are also referred to as HPV-negative tumors, but the proportion of human papillomavirus-related (HPV-positive) tumors is increasing worldwide [2]. Over the last several decades, it has become clear that these two tumor types are distinct entities with different biological and clinical characteristics [3,4,5].

Radiotherapy (RT) is one of the main treatment options for locally advanced HNC. In recent decades, RT delivery and planning have underwent immense technological advancements, which have reformed RT to be an effective and sophisticated tool for cancer treatment [1,2,6]. Although local or locoregional tumor control can be achieved in a large number of patients, there remains substantial room for improvement. This is especially the case for HPV-negative HNC patients where locoregional failure is an important cause of death [4,5,6,7,8].

Clinical factors, such as tumor and patient characteristics and the inaccurate delivery of RT doses can explain some of the local recurrences. However, it is commonly observed that tumors with the same clinical characteristics treated with the same treatment protocol respond differently. These differential responses can be due to various tumoral (e.g., HPV) and tumor microenvironmental factors (e.g., hypoxia). Hence, a better understanding of the interplay between therapy response, molecular tumor biology and the surrounding tissue is of utmost importance and will lead to improved treatment responses [7,9].

Preclinical models resembling human tumors are essential to understand the biological mechanisms of cancer formation and therapy response. In the field of radiation biology, clonogenic radiation survival curves, in which the number of surviving clones are measured, are the gold standard to determine the relative radiosensitivity of cells or to determine the effect of drug treatments [10]. This is because clonogenic survival assays are an efficient, practical and low-cost model [11,12,13,14].

However, these models lack the typical tumoral architecture, including cell–cell interactions and the tumor microenvironment (TME), making them poor representors of tumors and poor predictors of heterogenous treatment responses [11,13,15]. In vivo xenografts, especially patient-derived xenograft models, represent patient tumors more accurately compared to cell lines. However, they are associated with a significant burden of cost and infrastructure and have highly variable success rates [12,13]. Because of these drawbacks, 3D spheroid models have gained popularity in cancer research by bridging the gap between 2D and animal models. Sharing the simplicity and efficiency of 2D cultures, 3D models more accurately represent the dynamic tumoral architecture, including typical features such as naturally occurring hypoxic regions and cell–cell interactions [13,16]. Although these features are important factors in determining the RT response, the use of 3D models in the field of radiobiology is still limited [10].

In HNC especially, preclinical studies show higher radiosensitivity for HPV-positive HNC when compared to HPV-negative HNC [3,17]. However, data comparing the RT response of HPV-positive versus HPV-negative HNC in 3D spheroids to the existing gold-standard 2D and in vivo models are missing [15,18,19]. In this study, we validated the use of 3D spheroids in the RT response to HNC by comparing the RT responses of HPV-positive and HPV-negative 3D spheroids to those of 2D and in vivo models. Moreover, by performing whole-mount staining with the hypoxia marker pimonidazole and with the proliferation marker Ki-67, we assessed the potential use of 3D spheroids for radiobiological research in a spatial manner.

## 2. Results

### 2.1. HPV-Negative Spheroids Showed a Faster Growth Rate Compared to HPV-Positive Spheroids

To assess whether we could recapitulate the differential RT sensitivity of HPV-positive and HPV-negative HNC cells in 3D, the growth of four HNC cell lines in 3D conditions was investigated. Although morphological differences were noted, all four HNC cell lines formed 3D spheroids in ULA plates (Figure 1A). FADU cells formed compact spheroids. CAL27 cells also formed tight spheroids, but also showed the budding of several smaller cell aggregates. The HPV-positive SCC47 and SCC154 spheroids were characterized by compact aggregates surrounded by loose cells (Figure 1A), suggesting the influence of HPV infections on the extracellular matrix (ECM) organization.

Growth curves were generated by measuring the mean area of the spheroids (Figure 1B). All of the spheroids showed an increase in mean area over time after an initial phase of area reduction and stabilization for the CAL27, SCC47 and SCC154 spheroids. In general, HPV-negative spheroids showed a faster growth rate compared to HPV-positive spheroids, with a mean doubling time of 28 and 50 days, respectively.

### 2.2. The RT Response of HPV-Positive and HPV-Negative 3D Spheroids Correlated with 2D Clonogenic Assays

To determine whether 3D spheroids can be used for radiobiological studies, we assessed the RT response of 3D spheroids and compared it to the RT response determined by clonogenic survival assays.

The RT response of 3D spheroids was assessed by growth curves with a follow-up time ranging from 4 to 8 weeks after treatment. The RT doses of 0, 4 and 8 Gy were chosen based on our previous experience with in vitro and in vivo models of HNC [4,8]. For both HPV-positive and HPV-negative spheroids, a dose–response relationship could be detected, with the dose of 8 Gy showing a more effective inhibition of growth compared to the dose of 4 Gy (Figure 2A,B). For the HPV-negative spheroids (Figure 2A), a RT dose of 4 Gy resulted in a limited but significant reduction in the mean area of CAL27 spheroids from day 15 till day 42 (*p* = 0.049), and there was no reduction in the growth of the FADU spheroids compared to their respective non-RT-treated controls. After treatment with 8 Gy, the FADU spheroids showed an average reduction in area of 1.2-fold until day 19 post RT, whereas CAL27 showed an average reduction of 2.7-fold until day 28 (*p* = 0.019). These data indicate that the FADU spheroids showed a higher resistance to RT compared to the CAL27 spheroids.

For both the HPV-positive spheroids (Figure 2B), the RT dose of 8 Gy did not result in an increase in area during the 60-day observation period. Similar to the HPV-negative spheroids, differences within HPV-positive spheroids could be detected after treatment with 4 Gy (Figure 2B). The SCC47 spheroids showed only a limited growth stabilization until day 15 and showed regrowth on day 19. On the contrary, the SCC154 spheroids showed a growth stabilization until day 35 post-RT and did not reach regrowth during the follow-up on day 46 post-RT. When comparing the RT response of HPV-positive spheroids to HPV-negative spheroids, the HPV-positive spheroids showed an increased RT response, which was especially clear after treatment with a RT dose of 8 Gy (Figure 2A,B). This difference in response was seen in the reduction of the volumes compared to the untreated control spheroids, which showed, on average, a 7.6-fold decrease for HPV-positive spheroids and a 2-fold decrease for HPV-negative spheroids. In addition, the HPV-positive spheroids showed longer stabilization and/or inhibition of growth compared to the HPV-negative spheroids (Figure 2A,B), which was, on average, 46 days for HPV-positive spheroids and 23 days for HPV-negative spheroids. These results were in concordance with the results of the clonogenic assays, in which both HPV-positive (SCC154 and SCC47) HNC cell lines showed an increased RT sensitivity (*p* < 0.0001) compared to HPV-negative (FADU and CAL27) cell lines (Figure 3A). This recapitulation of the different RT sensitivity between HPV-positive and HPV-negative HNC in both 2D and 3D models indicates a good correlation between these two preclinical models.

### 2.3. The RT Response of HPV-Positive and HPV-Negative 3D Spheroids Correlated with In Vivo Data

Although 3D models, such as spheroids, have proven to be a tool to bridge the gap between 2D and in vivo models in precision medicine [10], in the radiobiology field, data comparing 3D in vitro models to in vivo models are limited. Hence, we assessed whether we could recapitulate the RT response seen in 3D spheroids (Figure 2A,B) in vivo by comparing the growth curves of HPV-negative CAL27 and HPV-positive SCC154 spheroids with in vivo tumor growth curves of the same cell lines (Figure 3B).

In line with the growth curves of the spheroid models, the tumor growth curves showed a prolonged RT response for HPV-positive SCC154 tumors and a more limited response in HPV-negative CAL27 tumors. More specifically, HPV-negative CAL27 tumors showed a stabilization in growth 8 days after treatment, and resulted in an average 1.8-fold difference. The HPV-positive SCC154 tumors showed an average tumor reduction of 3.5-fold starting 5 days after RT treatment, and a regrowth 31 days post-RT. All these data verify that 3D spheroids are able to capture the differential RT response of HPV-negative and HPV-positive HNC in correspondence with in vivo models.

### 2.4. Use of 3D Spheroids as a Preclinical Model to Study Mechanisms Underlying RT Response

The differential RT response within HPV-positive and HPV-negative 3D spheroid models (Figure 2A,B) illustrates the importance of complex preclinical models for understanding the heterogenous RT response seen in HNC patients. To investigate the possibilities of using 3D spheroids as a tool for studying radiobiological processes in a spatial manner, we performed, as a proof of concept, the whole-mount staining of proliferating and hypoxic cells at the end of their follow-up time. These are two important processes that have been known to negatively influence therapy response [3].

The proliferative capacity, measured as the mean Ki-67 staining intensity, of the spheroids was in accordance with the respective growth curves. Spheroids in the active growth phase at the end of the follow-up time, which mostly included HPV-positive spheroids, showed a significantly higher Ki-67 intensity (*p* < 0.0001) compared to spheroids in the plateau phase, which mostly included HPV-negative spheroids (Figure 4A). RT treatment did not influence the proliferation signal since the analyses were carried out at the end of the follow-up time. However, a non-significant (*p* = 0.33) difference in Ki-67 intensity was seen in RT-treated SCC154 spheroids compared to their untreated control spheroids. This corresponded with the exponential growth seen in the RT (4 Gy)-treated spheroids. It should be noted that we were not able to perform immunostaining on HPV-positive SCC154 and SCC47 spheroids treated with the RT dose of 8 Gy, since this dose resulted in the complete disintegration of the spheroids during the follow-up time.

Spheroids are known to be a good model for naturally occurring hypoxia [13]. We assessed the spatial distribution of hypoxia in HPV-positive and HPV-negative spheroids via pimonidazole (Hypoxyprobe) staining, which is an exogeneous hypoxia marker that captures the naturally occurring hypoxic regions in 3D spheroids. No differences in the amount of hypoxia were seen between the HPV-positive and HPV-negative spheroids (Figure 4B). However, in line with Ki-67 staining, a different distribution of pimonidazole-positive cells could be detected within different spheroids, which opens up opportunities to further investigate molecular processes on a spatial level.

## 3. Discussion

The introduction of 3D models, such as spheroids and organoids, in preclinical cancer research have resulted in the better recapitulation of the pathophysiology and heterogeneous biology of cancers, bridging the gap between 2D and in vivo models [13]. Despite these tremendous advantages and increased popularity in cancer research, the use of 3D models in the field of radiobiology is still limited.

Most studies on 3D HNC models have investigated molecular expression patterns and drug penetration [12,20], whereas the RT response of 3D HNC models has been studied less. For example, Kadletz et al. described the response of HPV-negative HNC spheroids to cisplatin and RT and compared this with the response in a 2D culture [15]. Zhang et al. used two new HPV-positive cell lines to compare their RT response to HPV-negative cell lines in 2D and 3D models [18], whereas Storch et al. compared the RT response in 3D spheroid, in vivo and 2D culture models of HPV-negative HNC [21]. Vitti et al. made use of spheroids to assess the response to proton and photon irradiations in combination with DNA repair inhibitors both in HPV-positive and HPV-negative HNC [22].

In this paper, we investigate the role of 3D spheroid models in the assessment of the RT response in HNC by comparing it to the gold-standard 2D clonogenic assays and to the tumor response in xenograft models. To the best of our knowledge, we are the first to compare the RT response of HPV-positive and HPV-negative HNC in 2D, 3D and in vivo models.

The differential RT response of HPV-positive and HPV-negative HNC is well documented [3,4,23,24,25,26], and therefore, used by us as a point of comparison. Previous studies reported an increased radioresistance in 3D models compared to 2D cultures [13,21,27,28,29]. Naturally occurring hypoxia in 3D spheroids certainly contributes to this resistance, since oxygen is needed to stabilize RT-induced DNA damage [30]. We anticipated this by using RT doses of 4 and 8 Gy. The latter is an RT dose we did not use in our clonogenic assays. Using these RT doses, we saw a dose-dependent response in the 3D spheroids that correlated with the radiosensitivity seen in 2D clonogenic assays.

Next, we compared the RT response of the spheroids with the tumor response in vivo via tumor growth curves. The growth curves of the spheroids showed a good correlation between HPV-positive SCC154 and HPV-negative CAL27 spheroids and their respective xenografts. However, the initial phase of the response curve of CAL27 differed between our 3D and in vivo models. In the CAL27 spheroids, an initial decrease in volume was seen in both the control and the RT groups, whereas in vivo, the tumors of both groups increased over the first 8 days. This discordancy between spheroids and xenografts can be explained by differences in growth in 3D culture and in vivo culture conditions. Additionally, the variety in RT doses given might play a role in the differences between different preclinical models, since we irradiated the spheroids with single doses of 4 and 8 Gy, whereas the xenograft received a fractioned regimen of 5 × 2 Gy. This fractionation scheme was chosen based on our previous experience with in vitro and in vivo models of HNC [4,8], but we are aware of the risk of biases for comparison when using different RT doses.

Despite the similarities in RT response between the 2D and 3D models, it should be noted that the in vitro RT sensitivity reflects the clonogenic survival fraction and the intrinsic RT sensitivity of the cell lines in 2D. It is well described that the DNA damage response and DNA repair capacity of the HNC cells play an important role herein [31]. In contrast, in vivo and spheroid models have a 3D architecture, and the intrinsic RT sensitivity is not the sole factor determining the RT response. Here, tumor microenvironmental factors such as hypoxia, intercellular interactions and spatial organization can have an influence on the RT response. In addition, 3D spheroids mimic the tumoral architecture, including cell–cell interactions and chemical/nutrient gradients, which are shown to be important determinants of the RT response [10,13]. In cancer research in general, the importance of the TME and its role in tumor progression and treatment responses has become a prime subject. The recent advances in single-cell and spatial multiomics analyses allow for studying the cellular components of tumors and their TME at the single-cell level in a spatial manner. Intratumoral heterogeneity, intercellular interactions and tumoral spatial organization are being revealed, shedding light on the mechanisms behind tumor progression and treatment resistance [32,33,34,35]. In line with this, several groups have incorporated compounds of the tumor microenvironment and extracellular matrix in 3D cultures, thereby mimicking the native TME in 3D models [36,37,38]. By using these 3D co-cultures for radiobiology experiments, and given the involvedness of the TME in determining the RT response of cancer cells, we can see the possibilities of using 3D models in radiobiological research [10,13,18].

Another advantage of using 3D spheroids lies in the field of radiosensitizing treatments, where drugs are given together with RT to enhance the RT effects. Although the lack of vasculature is an important drawback for these models, they recapitulate many other characteristics of in vivo tumors that are important in drug penetration, such as distinct cell layers and oxygen/nutrient gradients [13]. In line with this, we assessed the spatial distribution of oxygen gradients and proliferative capacity in our spheroids by pimonidazole and Ki-67 staining, respectively. Although we did not find differences in the amount of hypoxia and proliferative capacity between HPV-positive and HPV-negative spheroids, differences in the distribution of staining were noted. We hereby showed an additional advantage of 3D spheroids, since whole-mount staining provides important information about the spatial distribution and organization, and offers great opportunities regarding the assessment of molecular processes on a spatial level.

An important limitation of our study is the small number of cell lines that were tested, with two HPV-positive and two HPV-negative HNC cell lines used for the in vitro models and one HPV-positive and one HPV-negative HNC cell line used in vivo. It would also have been interesting to investigate the proliferative capacity (Ki-67 staining) and hypoxia level of the spheroids at different time points during the observation period and not only at the end. This would have allowed us to assess the influence of RT and regrowth on the proliferative capacity of different HNC cell lines. Despite the limitations, this proof of concept study shows an agreement between differences in the RT responses of HPV-positive and HPV-negative 2D and 3D models, including in vivo xenografts. It clearly shows the advantages and opportunities of the use of 3D spheroid models in the field of radiobiology, and can form a solid base for further 3D model-based research.

## 4. Materials and Methods

### 4.1. Cell Lines and Reagents

The HPV-positive HNC cell line SCC154 was purchased from the German collection of microorganisms and cell cultures (DSMZ, Braunschweig, Germany). HPV-positive HNC cell line SCC47 was gifted by Dr T. Carey, University of Michigan. Both HPV-positive cell lines were cultured in Minimal Essential Medium (MEM, Thermo Fisher Scientific, Waltham, MA, USA) and supplemented with 10% fetal bovine serum (FBS, Thermo Fisher Scientific, Waltham, MA, USA), 1% L-glutamine (Thermo Fisher Scientific, Waltham, MA, USA) and 1% non-essential amino acids (Thermo Fisher Scientific, Waltham, MA, USA). HPV-negative HNC cell lines FADU and CAL-27 were gifted by Dr. Mechthild Krause, Technical University Dresden, and by Dr. A. Begg, the Netherlands Cancer Institute, respectively, and were cultured in Dulbecco’s Modified Eagle Medium (DMEM, Thermo Fisher Scientific, Waltham, MA, USA), which was supplemented with 10% FBS and 1% sodium pyruvate (Thermo Fisher Scientific, Waltham, MA, USA). Cell lines were incubated at 37 °C and passaged via trypsinization. Cell lines were authenticated with short-tandem repeat profiling by the ATCC. All experiments were performed with mycoplasma-free cells.

### 4.2. Clonogenic Assay

Cells were seeded at different densities: 555, 1110, 2220 and 13,320 cells per well for the HPV-positive cell lines and 580, 870, 1740 and 5800 cells for the HPV-negative cell lines. They were exposed to increasing RT doses of 0, 2, 4 and 6 Gy. RT was delivered via the Small Animal Radiation Research Platform (SARRP, X-strahl, Camberley, UK). For the in vitro experiment, the following parameters were used: X-rays, 220 kV photons, 13 mA, a dose rate of 4 Gy/min, a source-to-skin distance (SSD) of 35 cm and a broad focus. Colonies were fixed after 2–3 weeks with 2.5% glutaraldehyde in PBS and stained with 0.4% crystal violet (Sigma-Aldrich, Saint-Louis, MO, USA). Colonies containing 50 cells or more were counted with Gelcount (Oxford Optronix, Abington, UK). The plating efficiencies were determined by dividing the amount of counted colonies by the amount of seeded cells. The survival fractions were calculated via normalization to the plating efficiency of non-irradiated controls of the respective cell lines, as previously described [39].

### 4.3. Spheroids

Spheroid formation was achieved by using ultra-low attachment (ULA) culture plates (96-well spheroid microplates with ULA surface; Corning, Costar, NY, USA). HPV-positive spheroids were grown in MEM supplemented with 20% FBS, 1% L-glutamine and 1% non-essential amino acids, and HPV-negative spheroids were grown in DMEM supplemented with 20% FBS and 1% sodium pyruvate. The seeding densities were 5000, 4000 and 1000 cells per well for the HPV-positive (SCC154 and SCC47), CAL27 and FADU spheroids, respectively. Plates were centrifuged at 1300 RPM for 10 min at room temperature. Spheroids were allowed to rest for 72 h and were cultured in a humidified 5% CO_2_ atmosphere at 37 °C. After 72 h of incubation, the medium was refreshed twice a week and cell growth was measured using an inverted widefield microscope (Olympus IX71, Olympus Corporation, Tokyo, Japan). Spheroids were treated with single-dose RT of varying doses (0, 4 and 8 Gy). HPV-negative spheroids were irradiated on day 7, whereas HPV-positive spheroids were irradiated on day 14 after seeding. These timepoints were determined based on the faster growth rate of HPV-negative spheroids.

### 4.4. In Vivo Xenograft Models

CAL27 cells (2 × 10^6^) or SCC154 cells (7.5 × 10^6^) were subcutaneously injected in each flank of 6–7-week-old female nu/nu Naval Medical Research Institute (NMRI) mice (Janvier Labs, France). The mice were either treated with a vehicle or RT (given in 5 fractions of 2 Gy, with a total dose of 10 Gy). RT was delivered via the Small Animal Radiation Research Platform (SARRP, X-strahl, Camberley, UK). Dose calculations were performed with MuriPlan software (X-strahl, Camberley, UK) after cone beam CT. The dose was delivered using 220 kV photons, 13 mA and a 10 × 10 mm collimator.

Tumor volumes (V = π/6xd_1xd_2xd_3) were determined with caliper measurements. The body weight and welfare of the mice were monitored daily during treatment and three times per week during follow-up. The experiment was performed according to the Ethical Committee for Animal Experimentation of KU Leuven (P163/2017; approval date 29 September 2017).

### 4.5. Immunofluorescence

After 35 (FADU) or 60 (CAL27, SCC154 and SCC47) days in culture, spheroids were incubated with 20 μg/mL of pimonidazole (1:1000, Hypoxyprobe-1, HP-1000, BioConnect) for 2 h at 37 °C, and the spheroids were fixed with 4% paraformaldehyde (PFA). After permeabilization with 0.1% Triton X-100 (Sigma-Aldrich, Saint-Louis, MO, USA) in 0.5% BSA-PBS, cells were stained with the mouse primary antibody against Ki-67 (1:200, 9449 CST, Danvers, MA, USA) or the rabbit primary antibody against pimonidazole (1:250, HypoxyProbe Omni Kit, HypoxyprobeTM, Burlington, MA, USA) for 24 h at 4 °C, followed by staining with goat anti-mouse 488 Alexa Fluor secondary antibody (1:500, 4408 CST, Danvers, MA, USA) or with goat anti-rabbit 647 Alexa Fluor secondary antibody (1:500, A21241, Invitrogen, Thermo Fisher Scientific, Waltham, MA, USA), respectively, for 24 h at 4 °C. Nuclei were counterstained with DAPI (1:500, D9542, Sigma-Aldrich, Saint-Louis, MO, USA). Immunofluorescence images were acquired with a confocal fluorescence microscope (NIKON C2, Nikon Corporation, Tokyo, Japan) and were quantified using Fiji (ImageJ).

### 4.6. Statistical Analysis

Statistical significance was tested by using the two-tailed *t*-test or two-way ANOVA in Graphpad prism 6.01 (GraphPad Prism Software Inc., San Diego, CA, USA). *p*-values < 0.05 were considered statistically significant.

## Figures and Tables

**Figure 1 ijms-24-03763-f001:**
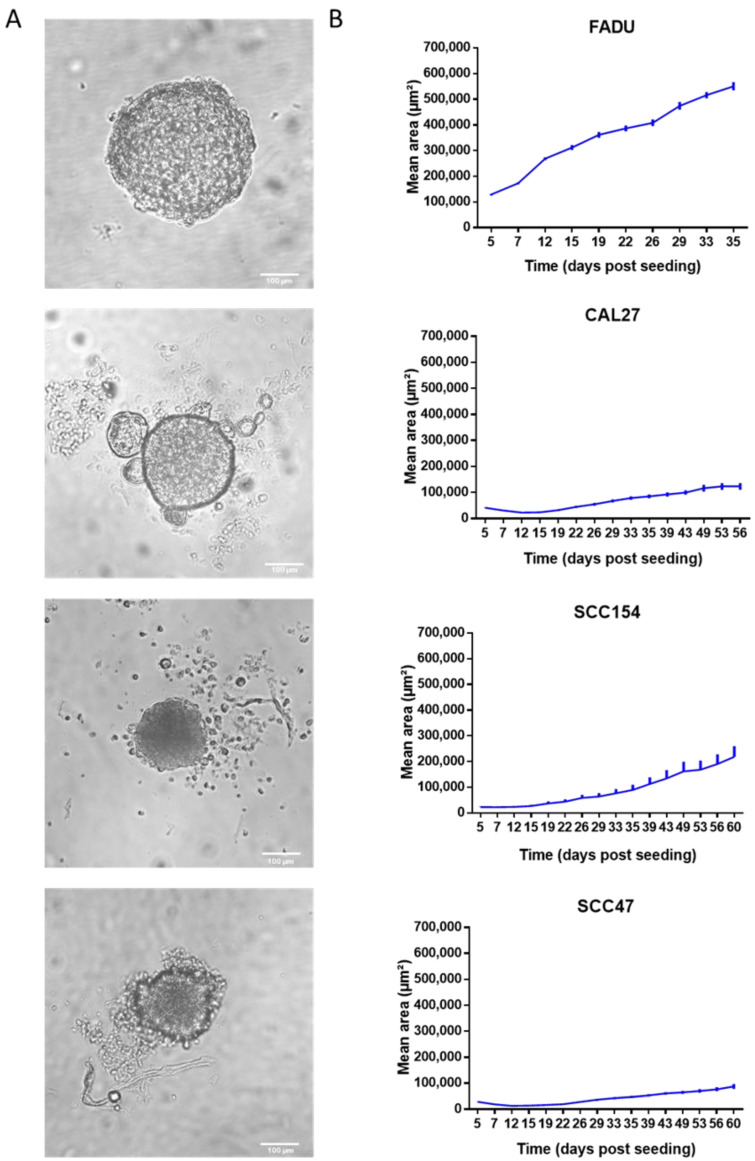
HPV-negative and HPV-positive HNC cells formed 3D spheroids. (**A**) Images of HPV-negative FADU and CAL27 spheroids and HPV-positive SCC154 and SCC47 spheroids taken on day 5. Scale bar, 100 μm. (**B**) Absolute tumor growth curves of HPV-negative FADU and CAL27 and HPV-positive SCC154 and SCC47 spheroids. Day 1 indicates the day of seeding. Data are presented as mean ± s.e.m., *n* = 3.

**Figure 2 ijms-24-03763-f002:**
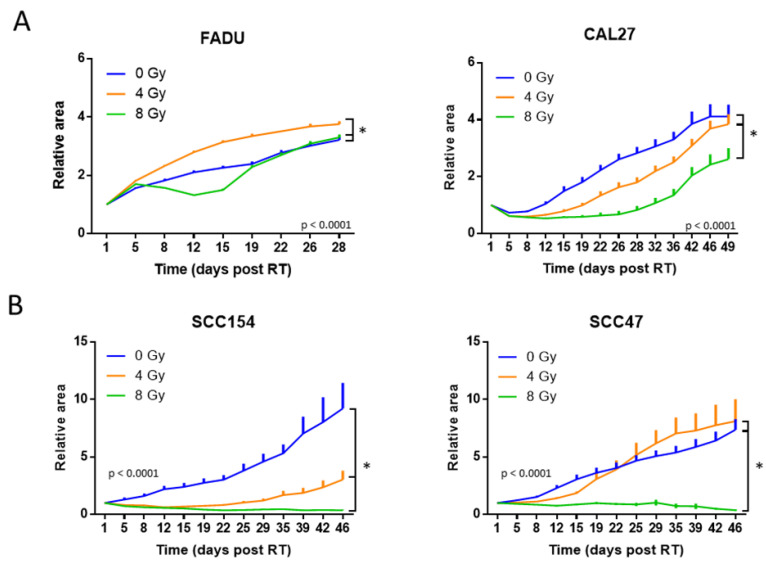
RT response in 3D spheroids of HPV-negative and HPV-positive cells assessed by tumor growth curves. (**A**) Relative tumor growth curves of HPV-negative FADU and CAL27 spheroids. The spheroids were treated with 4 Gy and 8 Gy at day 7 after seeding. (**B**) Relative tumor growth curves of HPV-positive SCC154 and SCC47 spheroids. The spheroids were treated with 4 Gy and 8 Gy at day 14 after seeding. (**A**,**B**) Relative curves were normalized to the spheroid area at the start of RT treatment. Day 1 indicates the start of treatment. Data are presented as mean ± s.e.m., *n = 3*. * *p*-values < 0.05 were determined by two-way ANOVA.

**Figure 3 ijms-24-03763-f003:**
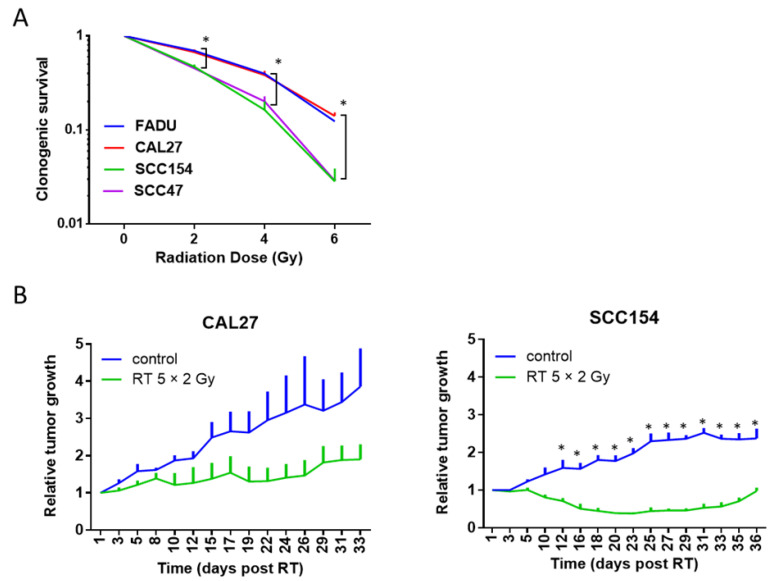
RT response of HPV-negative and HPV-positive cells assessed by clonogenic assays (**A**) and in vivo xenograft models (**B**). (**A**) HPV-negative (FADU and CAL27) and HPV-positive (SCC154 and SCC47) cells were irradiated with the indicated RT doses. Clonogenic cell survival is shown as the mean clonogenic survival fraction ± s.e.m., *n = 3*. (**B**) Relative tumor volume curves of HPV-negative CAL27 and HPV-positive SCC154 xenografts treated with vehicle with RT (5 × 2 Gy). Relative tumor growth curves were normalized to the tumor volume at the start of treatment day 1. Day 1 indicates the start of RT treatment. Data are presented as mean ± s.e.m., *n* = 3. (**A**,**B**) * *p*-values < 0.05 were determined by two-way ANOVA with Bonferroni correction for multiple testing.

**Figure 4 ijms-24-03763-f004:**
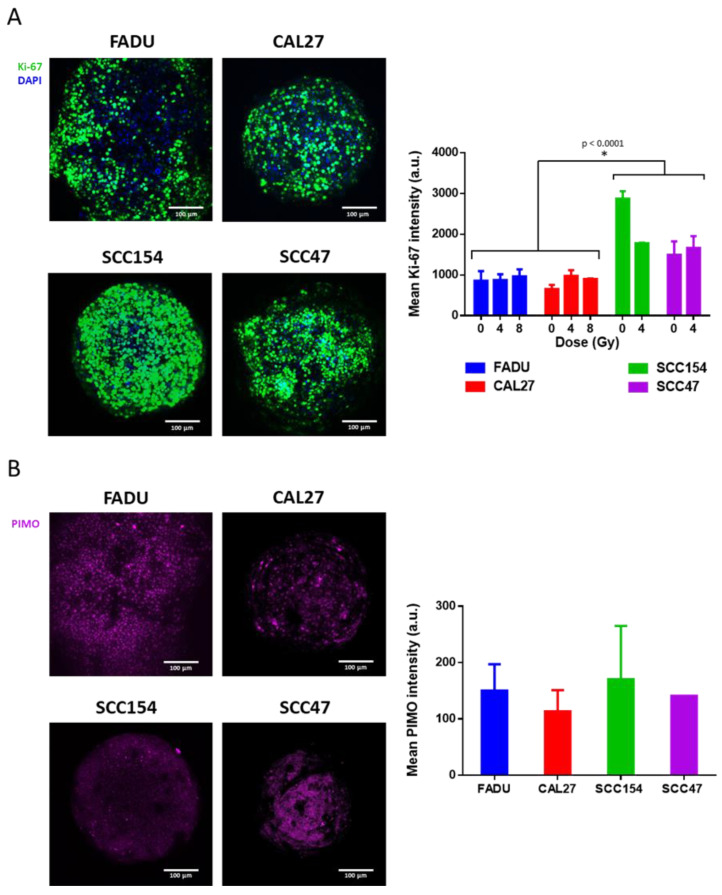
Proliferative capacity (**A**) and hypoxia levels (**B**) of HPV-negative and HPV-positive spheroids. (**A**) Whole-mount staining of Ki-67-positive cells (left) and mean intensity (right) in HPV-negative (FADU and CAL27) and HPV-positive (SCC154 and SCC47) spheroids at the end of the experiment. (**B**) Whole-mount staining of pimonidazole-positive cells (left) and mean intensity (right) in HPV-negative (FADU and CAL27) and HPV-positive (SCC154 and SCC47) spheroids. (**A**,**B**) Data are presented as mean ± s.e.m., *n =* 2 at minimum. *p*-values < 0.05 were determined by two-tailed *t*-test. Scale bar, 100 μm.

## Data Availability

The data presented in this study can be made available upon request from the corresponding author.

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
