# Peer review of "Use of 3D Spheroid Models for the Assessment of RT Response in Head and Neck Cancer"

_ijms, 2023, doi:10.3390/ijms24043763_

Round 1

Reviewer 1 Report

The manuscript ”Use of 3D spheroids models for the assessment of RT response of head and neck cancer” by Wegge et al., is a very important and interesting research work for head and neck cancer.

The results are well structured and the manuscript is easy to read/follow. I recommend the manuscript to be published in the journal with some minor corrections/suggestions below:

1)       Images of spheroids after RT for all 4 cell lines could be added to Figure.1 or as supplementary figure to get a better overview of the effect of RT.

2)      In Figure.3 please add the radiation dose next to RT (2Gy) in the figure itself. This makes it easier for the reader to follow

3)      Figure.4 – where 8 Gy used for SCC154 and SCC47 as well to check the Ki-67 intensity? What are the findings. Please specify

4)      In the material and methods section. Please add the cell densities for each cell line used for “clonogenic assays” and “spheroids”.  This makes it easier for data reproducibility

5)      The discussion could use a little bit more depth and comparison to other 3D studies in HNC or other 3D cancer studies

Reviewer 2 Report

In the present work the authors present a model of evaluating the effects of radiation therapy (RT) on tumor cell lines. Their work is very interesting and it has merit for publication.

The authors have done a good work presenting their data and conclusions. Their manuscript is well-written and well-structured. The effects of RT on tumors is a major issue in anti-tumor therapies and very little is known about its biology.

The HPV-negative cell lines, appear to recover from RT treatment (at all Gy), while the HPV-positive cell lines appear to manifest lower rates of recovery and at 8Gy not at all. Does this correlate with the observed ki-67 levels? please comment on that.

In addition (figure 3), in the in vivo experiments it appeared that cell death (as observed in clonogenic survival) reached levels below 10% for the HPV-positive cells and just above 10% for the HPV-negative cells. Yet, both cell types appear to recover from the treatment and start growing again. Please comment on that. What is the possible survival mechanism.

Overall, this is a well-written manuscript and it has merit for publication. The authors should consider rewriting some parts of their manuscript, since their work obtained a relatively high similarity index.
